# Effects of Two Kinds of Commercial Organic Fertilizers on Growth and Rhizosphere Soil Properties of Corn on New Reclamation Land

**DOI:** 10.3390/plants11192553

**Published:** 2022-09-28

**Authors:** Xuqing Li, Qiujun Lu, Dingyi Li, Daoze Wang, Xiaoxu Ren, Jianli Yan, Temoor Ahmed, Bin Li

**Affiliations:** 1Institute of Vegetable, Hangzhou Academy of Agricultural Sciences, Hangzhou 310024, China; 2Hangzhou Agricultural and Rural Affairs Guarantee Center, Hangzhou 310020, China; 3Department of Biological Environment, Material and Environmental College, Shanxi Jinzhong Institute of Technology, Jinzhong 030600, China; 4Rural Vitalization Service Center of Hangzhou, Hangzhou 310020, China; 5Institute of Biotechnology, Zhejiang University, Hangzhou 310058, China

**Keywords:** corn, new reclamation land, pig manure and mushroom residue organic fertilizer, sheep manure organic fertilizer, commercial compound fertilizer, soil property, microbiome

## Abstract

Due to the development of urbanization and industrialization, a large amount of cultivated land resources has been occupied, while new reclamation land could expand the supply of usable land for food security. Organic fertilizers, such as crop residues, biosolids, sheep manure, mushroom residue, and biogas liquid, have been considered as an effective amendment in immature soil to improve its quality. Recently, two kinds of commercial organic fertilizers, pig manure and mushroom residue organic fertilizer (PMMR-OF), and sheep manure organic fertilizer (SM-OF), have been more regularly applied in agriculture production. However, the information available on effect of the two kinds of fertilizers on plant growth and rhizosphere soil properties in immature field is very limited. In order to evaluate PMMR-OF and SM-OF on immature soil, the soil quality and microbial community structure of corn rhizosphere soil samples under the two kinds of organic fertilizers at different concentrations was investigated. The results revealed a significant difference between commercial organic fertilizers (especially SM-OF) and chemical compound fertilizers (CCF) in soil properties and microbial community structure. Indeed, compared with the control based on16S and ITS amplicon sequencing of soil microflora, SM-OF caused a 10.79–19.52%, 4.33–4.39%,and 14.58–29.29% increase in Proteobacteria, Actinobacteria, and Ascomycota, but a 5.82–20.58%, 0.53–24.06%, 10.87–16.79%, 2.69–10.50%, 44.90–59.24%, 8.88–10.98%, and 2.31–21.98% reduction in Acidobacteria, Gemmatimonadetes, Bacteroidetes, Verrucomicrobia, Basidiomycota, Mortierellomycota, and Chytridiomycota, respectively. CCF caused a 24.11%, 23.28%, 38.87%, 19.88%, 18.28%, and 13.89% reduction in Acidobacteria, Gemmatimonadetes, Bacteroidetes, Verrucomicrobia, Basidiomycota, Chytridiomycota, but a 22.77%, 41.28%, 7.88%, and 19.39% increase in Proteobacteria, Actinobacteria, Ascomycota, and Mortierellomycota, respectively. Furthermore, redundancy discriminant analysis of microbial communities and soil properties of PMMR-OF, SM-OF, CCF, and the control treatments indicated that the main variables of bacterial and fungal communities included organic matter content, available P, and available K. Overall, the results of this study revealed significant changes under different fertilizer conditions (PMMR-OF, SM-OF, CCF, under different concentrations) in microbiota and chemical properties of corn soil. Commercial organic fertilizers, particularly SM-OF, can be used as a good amendment for the new reclamation land.

## 1. Introduction

The innumerable population and limitation of available land resources are China’s national conditions. With the development of urbanization and industrialization, a large amount of cultivated land resources has been occupied, which has led to food supply challenges in China [1]. To meet the demand for land, the barren mountain land and forest land in the mountainous areas of Zhejiang Province, China, have been reclaimed for agricultural use. However, in most situations, the new reclamation land is not suitable for plant growth due to acidity, high gravel contents, poor nutrient contents, and very low content of organic matter [2,3]. For example, Li et al. [4] reported that compared to the normal yield (about 27,600 kg/ha) of sweet potato in mature soil, the yield of sweet potato is only about 15,000 kg/ha in reclamation soil. Hu et al. [5] also showed that the production capacity of new reclamation land is only 10–30% of the occupied cultivated land in Hangzhou city of Zhejiang Province, China. Therefore, new reclamation land had to be proposed both in quantity and quality. 

It is well known that all the factors affecting the soil’s properties, including physical, chemical, and biological properties, could impact on soil quality [6,7,8]. To meet the requirement of crops, inorganic fertilizers (such as nitrogen, phosphorus, potassium fertilizers, and commercial compound fertilizers) were widely used for a long time, which not only lead to soil acidification and nitrate accumulation but also destroyed the quality of crop products [9,10,11]. However, the soil suitability is highly associated with the content of the soil organic matter, and organic fertilizers can increase the organic nitrogen content and available inorganic phosphorus of soil [12,13]. The quality of new reclamation land can be effectively improved by increasing the use of organic fertilizer, such as crop residues, biosolids, sheep manure, mushroom residue, biogas liquid, and so on, which can not only modify soil physical and chemical properties, but can also exert a great influence on microbial communities [3,4,14]. 

Recently, two kinds of commercial organic fertilizers, including pig manure and mushroom residue organic fertilizer and sheep manure organic fertilizer, have been more regularly applied in agriculture production. However, there was very limited information on the effect the two kinds of fertilizers had on plant growth and rhizosphere soil properties in immature land. The objective of this study was to evaluate the effect of the two kinds of commercial organic fertilizers at different concentrations on corn growth, the rhizosphere soil properties, and microbial community structure in new reclamation land. In addition, we also examined the correlation of the soil properties and microbial community structure. This will provide a scientific basis to provide effective organic fertilizers for improving the soil quality of new reclamation lands in the future.

## 2. Materials and Methods

### 2.1. Experimental Design

This experiment was carried out on new reclamation land in Zhoukeng Village (29°38′4″ N; 119°4′24″ E, 309m) of Chun’an City, Zhejiang Province, China. The soil was mountain yellow mud and sandy loam, based on the soil classification system of the FAO-UNESCO, while its fertility grade (1–10, with 1 being the best) wasgrade 3 and class 6 based on the nutrient classification standard of the second soil survey in China. The top 20 cm soils had a pH of 5.78, with 9.61 g/kg of organic matter contents, 0.62 g/kg of total N, 4.54 mg/kg of available P, and 348.97 mg/kg of available K. 

The experiment consisted of eight different treatments through the application of two kinds of commercial organic fertilizers at different concentrations or chemical fertilizers on new reclamation land. Two kinds of commercial organic fertilizers, including pig manure and mushroom residue organic fertilizer (PMMR-OF, provided by Chun′an Shicheng Soil Fertilizer Co., Ltd., Hangzhou, China), and sheep manure organic fertilizer (SM-OF, provided by Hangzhou Nanwuzhuang Soil Fertilizer Co., Ltd., Hangzhou, China), were selected and used in this study. In detail, 0.90 kg, 1.35 kg, and 1.80 kg of PMMR-OF, and 0.75 kg, 1.05 kg, and 1.35 kg of SM-OF were applied to each m^2^ of the experimental field, respectively. Meanwhile, 0.075 kg of chemical compound fertilizer (CCF, N-P-K, 16-16-16, purchased from Shenzhen Batian Ecological Engineering Co., Ltd., Shenzhen, China), was applied to each m^2^ of the experimental field. The treatment without any fertilizer was used as the control. The information of each fertilizer used in this study is listed in Table 1. 

The field plot experiment was completely randomly designed in this study and was conducted for three years (from 2019 to 2021). The area of each plot was 20 m^2^, and the width, length, and depth of the plot were 330, 600, and 20 cm, respectively, and the planting density of corn was 30 × 20 cm. Each year, the top 0–20 cm soil of experimental field was mixed with different fertilizers before sowing. The seeds of corn (cultivar ‘Zhongdan 106′, provided by Chinese Academy of Agricultural Sciences, Beijing, China) were sown in the above-mentioned new reclamation land mixed with different fertilizers, and the treatment without any fertilizers was applied as the control. The experiment was carried out from May to August each year while the ripe ears of corn were harvested about 115 days after sowing. Each treatment had three replicates. 

### 2.2. Measure of Corn Production and Soil

About 115 days after sowing, the ripe ears of corn and soil samples were collected. In detail, when corn was collected each year, about 1.0 kg of fresh soil samples from the 5–20 cm rhizosphere soil of each plot were collected using the quartering method by a shovel with scale, and then passed through a 0.45-mm sieve to remove debris and roots. The weight of air-dried corn eras was measured by a digital scale (TCS-50, Shanghai hento Industrial Co., Ltd., Shanghai, China). The soil properties, including the pH, organic matter content, total N, available P, available K, exchangeable Ca, and exchangeable Mg, were detected after the samples were dried at room temperature, as described in [15]. In detail, the pH of the soil was measured at a soil/distilled water suspension ratio of 1:5 (g/mL) with a pH meter (FE28, MettlerToledo, Zurich, Switzerland); the content of organic matter was determined by the K_2_Cr_2_O_7_ oxidation external heating method; the content of total N was determined using an automatic Kjeldahl distillation-titration unit; the content of available P was determined by hydrochloric acid–ammonium fluoride extraction molybdenum–antimony anti–colorimetry; the available K was extracted by ammonium acetate, and the contents were determined by using a flame photometer; the exchangeable Ca and Mg were extracted by using ammonium acetate, and contents were determined by using an ice3500 atomic absorption spectrophotometer. All the treatment had three replicates.

### 2.3. Soil Genome Sequencing and Analysis

#### 2.3.1. Collection of Rhizosphere Samples and DNA Extraction

20 g rhizosphere soil of corn was sampled in 2021 (the third year). Genomic DNA of the soil samples was extracted by using the E.Z.N.ATM Mag–Bind Soil DNA Kit (OMEGA, Norcross, GA, USA) according to the manufacturer′s instructions. The quality of extracted DNA was determined by using a NanoDrop (ND-1000) spectrophotometer (ThermoFisher Scientific, Wilmington, DE, USA).

#### 2.3.2. Soil Genome Sequencing

The bacterial diversity was evaluated by the 16S rRNA V3–V4 region, amplified with the primers 341F (5′–CCTACGGGNGGCWGCAG–3′) and 805R (5′–GACTACHVGGGTATCTAATCC–3′) [16], whereas, for fungal diversity, the ITS1 and ITS2 region was amplified using the primers ITS1F (5′–CTTGGTCATTTAGGAAG TAA–3′) and ITS2 (5′–GCTGCGTTCTTCATTCGATGC–3′) [17], respectively. The components of the PCR included 12 μL ddH_2_O, 15 μL 2× Hieff^®^ Robust PCR Master Mix, 1 μL DNA template, 1 μL (10 μM) each forward and reverse primer. The PCR amplicons were purified with Vazyme VAHTSTM DNA clean beads (Vazyme, Nanjing, China). Afterward, amplicons with equal amounts were pooled and 2× 250 bp pair-end sequencing was accomplished through the Illumina MiSeq system (Sangon Biotech (Shanghai) Co., Ltd., Shanghai, China).

### 2.4. Statistical Analysis

The bioinformatics analysis of the microbiome was accomplished as our study before [18]. In brief, paired-end reads were preprocessed using Cutadapt V1.18 software [19,20]. Following primer sequence removal, clean reads were subjected to clustering to generate operational taxonomic units (OTUs) using VSEARCH software with a 97% similarity cutoff [21]. After selection of the representative read of each OTU using the QIIME package (v2020.06) [22], all representative reads were annotated and blasted against the SILVA (v123) and Unite database [23,24]. The OTUs was analyzed by Origin (v2021) and visualized in the bar graphs. The alpha diversity indices, including Chao1 and Shannon index, were visualized in the box plots. The analysis of beta diversity was carried out to observe the structural variation of rhizosphere soil microbiome across samples with principal coordinate analysis (PCA) [21]. The difference of rhizosphere soil microbiota between groups was determined by analysis of similarities, and linear discriminant analysis effect size (LeFSe) was carried out by using default parameters to observe the differentially abundant taxa between groups [22]. To investigate the impact of environmental factor (such as pH, organic matter content, total N, et al.) on microbial community structure, redundancy discriminant analysis (RDA) was carried out using Origin (v2021). Furthermore, the SPSS 16.0 software (SPSS Inc., Chicago, IL, USA) was used for all statistical analysis. The levels of significance (*p* < 0.05) of the main treatments and their interactions were calculated through one-way analysis of variance (ANOVA) after testing for normality and variance homogeneity by Student’s *t*-test.

## 3. Results and Discussion

### 3.1. Effects of Two Kinds of Commercial Organic Fertilizers on Corn Production

In order to examine the effect of two kinds of commercial organic fertilizers on corn production on new reclamation land, the dry weight of corn ears was evaluated about 115 days after sowing. The result from this study indicated that the weight of air-dried corn ears was affected by two kinds of commercial organic fertilizers, while the effect was dependent on the kind and concentration of fertilizer. In detail, compared to the control, there was a significant increase in the weight of air-dried corn ears by PMMR-OF (12.23–79.90%, 19.17–81.38%, and 29.14–77.16% at 0.90, 1.35, and 1.80 kg/m^2^, respectively), SM-OF (14.58–78.29%, 15.46–80.52%, and 23.63–61.87% at 0.75, 1.05, and 1.35 kg/m^2^, respectively), and CCF (10.03–45.44%) from 2019 to 2021, respectively. However, there was no significant difference in the weight of air-dried corn ears between PMMR-OF and SM-OF treatments (Table 2).The various effects of PMMR-OF, SM-OF, and CCF on the biomass of corn ears may be due to the difference in their components.

Consistent with the results of this study, some previous studies have also reported the use of soil amendments in the production of crop. For example, Song et al. [25] reported that the application of inorganic nitrogen fertilizer, balanced inorganic fertilizers (NPK), NPK plus corn straw, and NKP plus farmyard manure resulted in significant increases in corn yield, but only organic matter amendments sustained increasing yield trends. Nedunchezhiyan et al. [26] reported that the application of rice straw and farmyard manure increased sweet potato growth and root yield under lowland conditions. Li et al. [27] revealed that sheep manure exhibited a great promotion effect on corn production in newly reclaimed land. On the other hand, in order to elucidate the effect of years on decomposition of organic residue and major traits, the agronomic data in the study should be further analyzed by giving analysis of variance for all traits, including treatment and year, and interaction of treatments and year effects.

### 3.2. Effects of Two Kinds of Commercial Organic Fertilizers on Soil pH, Organic Matter Contents, and Nutrient Elements

The results from this study indicated that there was no significant difference in soil pH between the addition of commercial organic fertilizer, chemical compound fertilizer, and the control of corn rhizosphere soil during 2019 to 2021. However, compared with the control, the soil pH was slightly increased by PMMR-OF, SM-OF, and CCF from pH 5.59 to pH 5.61–6.64 in 2019–2021, excluding a slight decrease by SM-OF to pH 5.40–5.54 in 2019, which is in coordination with our previous studies [13,17,27,28,29], and indicated that the pH of soil can be changed to a certain extent. Furthermore, the soil OMC was significantly increased by all the test commercial organic fertilizers in 2019–2021. Indeed, in 2019 and 2020, PMMR-OF and SM-OF caused a 24.63–39.61%, 25.39–50.71% increase in the soil OMC compared to the control, respectively, while the greatest increase was achieved by PMMR-OF at 1.80 kg/m^2^ (39.61%, greater than CCF′s 7.45% in 2019; 50.71%, greater than CCF′s 2.34% in 2020; respectively). PMMR-OF and SM-OF caused a 46.51–75.21% increase in the soil OMC compared to the control in 2021, while the greatest increase was achieved by SM-OF at 1.35 kg/m^2^ (75.21%, great than CCF′s 11.14%), and there was no significant difference in the OMC between CCF and the control in 2019–2021 (Table 3). The results from this study revealed that the soil OMC can be improved by commercial organic fertilizer, and SM-OF was slightly better than PMMR-OF but was unaffected by CCF. This may be due to the proportion of microaggregates (<0.25 mm) of soil decreased by repeat application of CCF [30]. In agreement with the results of this study, a similar observation of an increase in OMC was observed by Renet al. [17] and Li et al. [4,27]. Furthermore, other studies have also shown that the soil OMC could be effectively improved by planting winter legume crops, returning crop stems and stalks to the field, comprehensively utilizing agricultural and animal husbandry wastes, and making compost by using rural litter and gully soil [28,31,32,33,34,35,36,37]. Therefore, it can be inferred that commercial organic fertilizers have great potential to improve soil quality of the new reclamation soil.

Compared to the control, the total N was significantly increased by PMMR-OF, SM-OF, and CCF by 9.38–15.63%, 15.63–21.35%, and 13.54% in 2019, 27.32–29.27%, 20.98–30.73%, 8.78% in 2020, and 42.79–52.40%, 50.48–52.40%, 16.35% in 2021, respectively. The greatest increase was achieved by PMMR-OF at 1.80 kg/m^2^ and SM-OF at 1.35 kg/m^2^ (both by 52.40%, greater than CCF′s 16.35%) in 2021. Compared to the control, the content of available P was significantly increased by all the treatments, with a 52.91–73.19% increase in 2019, a 118.40180.00% increase in 2020, and a 230.00–258.40% increase in 2021, respectively. In detail, PMMR-OF at 1.35 kg/m^2^, SM-OF at 1.35 kg/m^2^, and SM-OF at 0.75 k/m^2^ caused greatest increase in available P in 2019, 2020, and 2021, which caused a 10.20%, 28.22%, and 6.37% increase when compared to CCF, respectively. Similarly, compared to the control, the content of available K was significantly increased (5.55–25.95%, 14.74–21.78%, 25.90–35.43% in 2019, 2020, 2021, respectively) by all the treatments, and there was significant increase by PMMR-OF, SM-OF, and CCF (17.84–22.83%, 5.55–16.03%, 25.95% in 2019, 17.86–19.47%, 14.74–19.11%, 21.78% in 2020, 28.63–33.36%, 28.35–35.43%, 25.90% in 2021, respectively). Results from this study revealed that the total N, available P was more increased by SM-OF than PMMR-OF, while there was no significant difference of available K between PMMR-OF and SM-OF in 2019, 2020, and 2021 (Table 4). Similar observations were found in previous studies. For example, Li et al. [3] found that the total N was significantly changed by mushroom residue, biogas liquid, and vegetable cake, with a 11.6–48.14% increase. Criquet and Braud [38] showed that P availability could be improved by sewage sludge and sewage sludge compost in the degraded Mediterranean soil. Li et al. [4] found that the available K was significantly increased by commercial organic fertilizer, sheep manure, and mushroom residue, with a 2.6–11.2% increase. However, the effect was dependent on the kind and concentration of organic fertilizers as well as the application time. 

Compared to the control, the exchangeable Ca was significantly increased by PMMR-OF (14.71–25.84%), SM-OF (11.87–16.62%, except at 0.75 kg/m^2^ slightly reduced by 3.19%), and slightly increased by CCF (6.25%) in 2019. There was a significant reduction by SM-OF at 0.75 kg/m^2^ (15.86%), a slight reduction by CCF (8.76%), a slight increase by SM-OF at 1.05 kg/m^2^ (9.37%), and there was no significant change by the other treatments in 2020. In 2021, the exchangeable Ca was reduced by all the treatments by 1.88–16.33% (except PMMR-OF at 0.90 kg/m^2^ with a slight increase by 2.88%), and the greatest decrease was achieved by PMMR-OF at 1.80 kg/m^2^ (16.33%). Similarly, compared to the control, the exchangeable Mg was slightly increased by PMMR-OF (2.57–5.21%), SM-OF (2.39–6.12%), and slightly reduced by CCF (0.61%) in 2019, and it was reduced by all the treatments by 2.65–12.06% in 2020, in which the greatest decrease was achieved by SM-OF at 0.75 kg/m^2^ (12.06%) in 2020. In 2021, the exchangeable Mg was slightly reduced by PMMR-OF at 0.90 kg/m^2^ (4.47%) and 1.35 kg/m^2^ (0.71%), and significantly increased by PMMR-OF at 1.80 kg/m^2^ (10.12%), SM-OF (17.41–19.76%), and CCF (25.18%), respectively (Table 5). The different effects of PMMR-OF, SM-OF, and CCF on exchangeable Ca and Mg may be related to the differences in their components and concentrations. 

Fertilizer application plays an important role in soil quality improvement and crop productivity enhancement [39,40]. For example, Li et al. [4] indicated that the application of organic fertilizer could improve the fertility and quality of the immature soil and promote the plant growth in the immature soil by adjusting the pH and OMC of soil, which has been regarded as the basis for soil fertility and quality. Li et al. [3] also reported that biological organic fertilizers couldeffectively improve the content of OMC, available P, and total N in newly reclaimed land. The results of this study showed that the soil pH, OMC, total N, available P, available K, exchangeable Ca, and exchangeable Mg were differentially affected by fertilizers. The effect was dependent on the soil nutrient elements, and the kind and concentration of fertilizers (Appendix A). In conclusion, after three years of continuous use of two kinds of commercial organic fertilizers, there was a greater increase by SM-OF than PMMR-OF in the soil pH and the content of OMC, total N, available K, and exchangeable Mg in soil. The content of available P was more greatly increased by PMMR-OF than by SM-OF, and the exchangeable Ca was less reduced by SM-OF than by PMMR-OF. Furthermore, it is well known that the organic matter amendments, such as farmyard manure and crop straw, cannot only increase crop yield but can also maintain a sustainable increasing trend in crop yield owing to the enhancement of soil fertility [31], and the organic matter has been proposed to exhibit a greater effect on microbial communities compared to the other parameters [15,41,42]. Therefore, the greater effect of SM-OF than PMMR-OF in improvement of new reclamation soil may be able to be inferred in this study. 

### 3.3. The Effect of Two Kinds of Commercial Organic Fertilizers in Microbial Community Diversity

The number of OTUs is shown in Figure 1. Results indicated that the average number of bacterial OTUs was 2975.33 (2880 to 3103), 2874.00 (2796 to 2925), 2888.00 (2814 to 3014), 2775.67 (2688 to 2851), 3022.00 (2934 to 3106), 2833.67 (2741 to 2886), 2865.00 (2796 to 2945), and 2819.33 (2698 to 2912) in PMMR–OF at 0.90, 1.35, 1.80 kg/m^2^, SM–OF at 0.75, 1.05, 1.35 kg/m^2^, CCF at 0.075 kg/m^2^ and the control, respectively. In detail, compared with the control, the bacterial OTUs number was slightly increased (5.53%, 1.94%, 2.44%, 7.19%, 0.51%, 1.62%, respectively) by PMMR–OF at 0.90, 1.35, 1.80 kg/m^2^, SM–OF at 1.05, 1.35 kg/m^2^, CCF at 0.075 kg/m^2^, and slightly reduced (1.55%) by SM–COF at 0.75 kg/m^2^, respectively. Meanwhile, the average number of fungal OTUs was 713.00 (703 to 727), 686.33 (625 to 735), 629.00 (610 to 659), 637.67 (603 to 666), 711.00 (657 to 752), 642.33 (563 to 691), 766.00 (691 to 814), and 596.67 (493 to 653) in PMMR–OF at 0.90, 1.35, 1.80 kg/m^2^, SM–OF at 0.75, 1.05, 1.35 kg/m^2^, CCF at 0.075 kg/m^2^ and the control, respectively. In general, PMMR–OF at 0.90, 1.35 kg/m^2^, SM–OF at 1.05 kg/m^2^, and CCF at 0.075 kg/m^2^ caused a significant increase (19.50%, 15.03%, 19.16%, 28.38%, respectively), and PMMR–OF at 1.80 kg/m^2^, SM–OF at 0.75, 1.35 kg/m^2^ caused a slight increase compared to the control, respectively. This is possibly due to the nutrients of the amendment soil being more suitable for bacterial growth than fungal growth.

Furthermore, the index of Chao1 and Shannon in the V3 + V4 region (bacteria) and ITS region (fungi) is shown in Figure 2 and Figure 3. The average Chao1 index of bacteria was 3865.59 (3804.36 to 3987.70), 3783.77 (3680.04 to 3870.81), 3738.89 (3678.06 to 3814.16), 3599.29 (3561.00 to 3628.05), 3969.29 (3912.66 to 4041.57), 3735.13 (3637.62 to 3800.41), 3739.48 (3694.15 to 3801.78), 3636.41 (3573.8 to 3697.26), and the average Shannon index of bacteria was 6.19 (5.90 to 6.44), 6.28 (6.20 to 6.39), 6.36 (6.15 to 6.49), 6.09 (5.83 to 6.26), 6.34 (6.18 to 6.49), 6.36 (6.27 to 6.50), 6.06 (5.79 to 6.21), and 6.26 (6.12 to 6.41) in PMMR–OF at 0.90, 1.35, 1.80 kg/m^2^, SM–OF at 0.75, 1.05, 1.35 kg/m^2^, CCF at 0.075 kg/m^2^ and the control, respectively. In general, the bacterial Chao1 index was increased (6.30%, 4.05%, 2.82%, 9.15%, 2.71%, 2.83%, respectively) by PMMR–OF at 0.90, 1.35, 1.80 kg/m^2^, SM–OF at 1.05, 1.35 kg/m^2^, CCF at 0.075 kg/m^2^ and slightly reduced (1.02%) by SM–OF at 0.75 kg/m^2^ compared with the control, respectively, among which PMMR–OF at 0.90 kg/m^2^ and SM–OF at 1.05 kg/m^2^ caused a significantly increase (6.30%, 9.15%) (Figure 2a), but no significant difference was observed in the Shannon index of bacterial community among all the eight treatments (Figure 2b). 

Meanwhile, the average Chao1 index of fungi was 831.63 (796.04 to 881.44), 827.81 (745.12 to 870.04), 749.22 (701.97 to 794.67), 748.48 (714.84 to 787.41), 820.37 (727.56 to 876.45), 732.97 (636.50 to 812.00), 927.33 (838.01 to 990.63), 701.93 (622.00 to 747.72), and the Shannon index of fungi was 4.07 (3.79 to 4.21), 3.66 (3.50 to 3.84), 4.07 (3.93 to 4.26), 3.88 (3.53 to 4.11), 4.06 (3.64 to 4.35), 4.34 (4.14 to 4.52), 3.98 (3.68 to 4.20), 4.22 (3.97 to 4.37) in PMMR–OF at 0.90, 1.35, 1.80 kg/m^2^, SM–OF at 0.75, 1.05, 1.35 kg/m^2^, CCF at 0.075 kg/m^2^ and the control, respectively. In detail, the fungal Chao1 index was increased (18.48%, 17.93%, 6.74%, 6.63%, 16.87%, 4.42%, 32.11%, respectively) by PMMR–OF at 0.90, 1.35, 1.80 kg/m^2^, SM–OF at 0.75, 1.05, 1.35 kg/m^2^, CCF at 0.075 kg/m^2^ compared with the control, respectively, among which CCF at 0.075 kg/m^2^ caused a significant increase (32.11%) (Figure 3a). The Shannon index of fungi was slightly reduced by PMMR–OF at 0.90, 1.35, 1.80 kg/m^2^, SM–OF at 0.75, 1.05 kg/m^2^, CCF at 0.075 kg/m^2^ (3.73%, 13.26%, 3.56%, 8.20%, 3.82%, 5.86%, respectively), among which PMMR–OF at 1.35 kg/m^2^ caused a significantly reduction (13.26%), and SM–OF at 1.05 kg/m^2^ caused a slight increase (2.78%) compared with the control (Figure 3b).

Furthermore, the average bacterial OTUs were 4.17-, 4.19-, 4.59-, 4.35-, 4.25-, 4.41-, 3.74-, 4.73-fold greater than those of fungi, the bacteria Chao1 indexes were 4.65-, 4.57-, 4.99-, 4.81-, 4.84-, 5.10-, 4.03-, 5.18-fold greater than those of fungi, the bacteria Shannon indexes were 1.52-, 1.72-, 1.56-, 1.57-, 1.56-, 1.47-, 1.52-, 1.48-fold greater than those of fungi in PMMR–OF at 0.90, 1.35, 1.80 kg/m^2^, SM–OF at 0.75, 1.05, 1.35 kg/m^2^, CCF at 0.075 kg/m^2^ and the control, respectively. In addition, PMMR–OF at 1.80 kg/m^2^ and SM–OF at 1.35 kg/m^2^ caused greater ratio of bacterial and fungal OUT distribution and Chao1 indexes compared to all the other treatments. The results indicated that the diversity of bacteria in the rhizosphere soil of corn was significantly increased and PMMR–OF at 1.80 kg/m^2^ and SM–OF at 1.35 kg/m^2^ significantly reduced the diversity of fungi.

Obviously, the bacterial and fungal diversity were differentially affected by different fertilizers. The role of organic fertilizer in soil microbe has been reported in many previous studies [3,4]. For example, commercial organic fertilizer, sheep manure, and mushroom residue caused changes in the number of OTUs, Chao1, and Shannon index in the microbial community structure compared with the control [4], while biogas slurry, vegetable cake, and mushroom caused a greater ratio of bacterial and fungal OTUs distribution and diversity indexes in immature soil [3]. According to previous studies, the soil fertility is associated with the richness of some soil microbes, which have the capacity to enhance the solubilization of insoluble phosphate compounds, produce IAA and siderophore, fix nitrogen, and be involved in a series of soil biological processes [43,44,45,46]. In other words, the improvement of the soil quality may be partially attributedto the enrichment of specific microbes.

### 3.4. The Effect of Two Kinds of Commercial Organic Fertilizers in Soil Microbial Community Structure

Principal component analysis (PCA) of the bacterial community structure indicated that the four replicates of PMMR–OF, SM–OF, CCF, and the control were divided into eight different groups; there was overlap among all the PMMR–OF, SM–OF, CCF treatment, and the control. However, the treatment PMMR–OF at 1.80 kg/m^2^ and SM–OF at 1.35 kg/m^2^ were well separated from CCF, indicating that the bacterial community structure of the rhizosphere soil was slightly changed by both PMMR–OF at 1.80 kg/m^2^ and SM–OF at 1.35 kg/m^2^ (Figure 4a). Similarly, PCA analysis of the fungal community structure indicated that the four replicates of each treatment were divided into eight different groups, while the treatment PMMR–OF at 0.90 kg/m^2^ and CCF were well separated from the control, and the treatment PMMR–OF at 1.35 kg/m^2^ was well separated from CCF, which indicates that the fungal community structure of rhizosphere soil was significantly changed by these PMMR–OF and CCF treatments (Figure 4b). In general, greater diversity was observed in the four replicates of PMMR–OF (1.80 kg/m^2^) and SM–OF (1.35 kg/m^2^) in the bacterial community structure, and greater diversity was also observed in the four replicates of PMMR–OF (0.90 kg/m^2^ and 1.35 kg/m^2^) in the fungal community structure, respectively (Figure 4). In agreement with the results of this study, continuous application of bio-organic fertilizer, commercial organic fertilizer, and microbial fertilizer caused a significant change in bacterial and fungal communities in barberry, cotton, and corn soil [27,47].

This result indicated that the application of PMMR–OF, SM–OF at different concentrations, and CCF resulted in a significant change in the composition of the bacterial and fungal community at the phylum (Figure 5) and genus levels compared to the control (Figure 6). Indeed, the top ten phyla in the rhizosphere soil of corn were selected to generate a relative abundance histogram, in which Proteobacteria, Acidobacteria, Gemmatimonadetes, Actinobacteria, Bacteroidetes, and Verrucomicrobia were the main bacterial phylum with a relative abundance of 36.90–45.31%, 15.01–20.63%, 6.68–9.45%, 4.02–8.34%, 3.54–7.89%, and 3.02–4.77%, respectively (Figure 5a). Furthermore, the relative abundance histogram based on the top 15 species showed that *Gemmatimonas*, *Gp6*, *Sphingomonas*, *Gp1*, *Gp3*, *GP4*, *WPS-1*, *Spartobacteria*, and *Subdivision3* were the main bacterial genes (average relative abundance > 1%) (Figure 6a). Compared with that of the control, the relative abundance of *Gemmatimonas* was reduced by 11.20%, 16.38%, 1.19%, 24.06%, 0.53%, and 23.28% in PMMR–OF (0.90, 1.35, 1.80 kg/m^2^), SM–OF (0.75, 1.05 kg/m^2^), and CCF, but increased by 7.49% in SM–OF (1.35 kg/m^2^), respectively. The relative abundance of *Gp6* was increased by 10.86%, 24.52%, 14.38%, and 34.19% in PMMR–OF (1.35, 1/80 kg/m^2^), SM–OF (1.05, 1.35 kg/m^2^), but reduced by 18.54%, 26.27%, and 42.65% in PMMR–OF (0.90 kg/m^2^), SM–OF (0.75 kg/m^2^), and CCF, respectively. The relative abundance of *Sphingomonas* was increased by 27.77%, 55.72%, 49.06%, and 1.52% in PMMR–OF (0.90, 1.35, 1.80 kg/m^2^), SM–OF (1.05 kg/m^2^), but reduced by 3.82%, 13.04%, and 7.85% in SM–OF (0.75, 1.35 kg/m^2^), CCF, respectively. The relative abundance of *GP1* was reduced by 55.73%, 58.11%, 75.85%, 44.63%, 73.87, and 20.54% in PMMR–OF (0.90, 1.35, 1.80 kg/m^2^), SM–OF (1.05, 1.35 kg/m^2^), CCF, but increased by 26.11% in SM–OF (0.75 kg/m^2^), respectively. The relative abundance of *GP3* was reduced by 28.17%, 42.32%, 39.66%, 12.81%, 29.80%, 37.01%, 11.35% in PMMR–OF (0.90 kg/m^2^), PMMR–OF (1.35 kg/m^2^), PMMR–OF (1.80 kg/m^2^), SM–OF (0.75 kg/m^2^), SM–OF (1.05 kg/m^2^), SM–OF (1.35 kg/m^2^), CCF, respectively. The relative abundance of *GP4* was reduced by 20.83%, 12.93%, 30.60%, 23.29%, and 54.47% in PMMR–OF (0.90, 1.35 kg/m^2^), SM–OF (0.75, 1.05 kg/m^2^), and CCF, but was increased by 20.06%, 28.26% in PMMR–OF (1.80 kg/m^2^), SM–OF (1.35 kg/m^2^), respectively. The relative abundance of *WPS-1* was increased by 34.36%, 34.70%, 15.61%, 5.52%, 2.48%, 6.62% in PMMR–OF (0.90, 1.35, 1.80 kg/m^2^), SM–OF (0.75, 1.05 kg/m^2^), CCF, but was reduced by 5.32% in SM–OF (1.35 kg/m^2^), respectively. The relative abundance of *Spartobacteria* was reduced by 14.45%, 32.29%, 25.94%, 5.53%, 7.45%, and 20.12% in PMMR–OF (0.90, 1.35, 1.80 kg/m^2^), SM–OF (0.75, 1.05 kg/m^2^), CCF, but was increased by 18.98% in SM–OF (1.35 kg/m^2^), respectively. The relative abundance of *Subdivision3* was reduced by 11.07%, 12.49%, 17.26%, and 17.03% in PMMR–OF (0.90, 1.35 kg/m^2^), SM–OF (1.05 kg/m^2^), and CCF, but was increased by 19.03%, 8.00%, and 33.46% in PMMR–OF (1.80 kg/m^2^) and SM–OF (0.75, 1.35 kg/m^2^), respectively (Figure 6a). The change in the number of specific bacteria may be mainly due to the different nutrients between different fertilizer treatments. In particular, more attention should be paid to *Gemmatimonas* and *Sphingomonas*. *Gemmatimonas* is reported to be associated with soil microbiome stability by phosphonate and phosphinate metabolism [48], and is capable of reducing the potent greenhouse gas N_2_O under both anaerobic and aerobic conditions [49,50]. *Sphingomonas* is reported to play a role in plant growth by producing plant growth hormones, e.g., indole acetic acid and gibberellins during such as drought, salinity, and heavy metal stress conditions in agricultural soil [51].

Meanwhile, according to the distribution and relative abundances of fungi in corn rhizosphere soil at the phylum level, Ascomycota, Basidiomycota, Mortierellomycota, and Chytridiomycota were the main fungal phyla, with relative abundance of 54.35–74.87%, 9.37–29.35%, 5.33–11.34%, 2.46–4.98%, respectively (Figure 5b). Furthermore, at the genus level, *Ramophialophora*, *Mortierella*, *Gibberella*, and *Aureobasidium* were the main fungal genera (average relative abundance > 1%) (Figure 6b). Compared with the control, the PMMR–OF (0.90, 1.35, 1.80 kg/m^2^), SM–OF (0.75, 1.05, 1.35 kg/m^2^), and CCF caused an 870.45%, 1096.40%, 1080.35%, 9.28%, 271.57%, 297.27%, 60.70% increase in the relative abundance of *Ramophialophora*, and a 21.28%, 48.05%, 70.57%, 25.58%, 3.84%, 45.88%, 22.37% reduction in the relative abundance of *Aureobasidium*. Interestingly, the relative abundance of *Mortierella* was reduced by 27.67%, 44.35%, 36.63%, 9.98%, 8.92% in PMMR–OF (0.90, 1.35, 1.80 kg/m^2^), and SM–OF (0.75, 1.05 kg/m^2^), but increased by 17.11% and 19.58% in SM–COF (1.35 kg/m^2^) and CCF, respectively. The relative abundance of *Gibberella* was reduced by 6.03%, 3.76%, 17.59%, 27.11%, 4.64%, and 5.29% in PMMR–OF (0.90, 1.35, 1.80 kg/m^2^), SM–OF (0.75, 1.05, 1.35 kg/m^2^), respectively, but increased by 17.58% in CCF (Figure 6b). The role of the anamorphic genus *Ramophialophora* in soil is still unclear. However, some strains of *Aureobasidium* can be used as a biocontrol agent for fire blight protection in organic apple and pear production [52]. Furthermore, studies have revealed that the fungi from the genus *Gibberella* have been reported to be the pathogen of many leaf- and soil-borne diseases [53]. However, *Mortierella* in the rhizophere of wheat is significantly associated with increasing crop growth and yield, and the increase in relative abundance of *Mortierella* may promote plant growth [34]. The conflict effect on the relative abundance of *Mortierella* and *Gibberella* revealed the complexity of the soil fungal community. In agreement with the results of this study, our previous studies also showed that the improvement of organic fertilizers in the soil quality may be mainly due to the change in the genera abundance distribution by enriching specific soil microbe in the newly cultivated land [3,4].

### 3.5. The Effect of Two Kinds of Commercial Organic Fertilizers in the Rhizosphere Microbiome and Biomarker

The linear discriminant analysis of effect size (LEfSe) (LDA > 4, *p* < 0.05) was used to reveal the biomarkers with the largest difference between the rhizosphere soil microbial communities of corn under two kinds of commercial organic fertilizers at different concentrations (Figure 7a). A total of 15 bacterial biomarkers were found in the PMMR–OF, SM–OF, CCF treatments, and the control, and SM–OF at 1.35kg/m^2^ identified nine types of bacteria. In detail, PMMR–OF at 1.35 kg/m^2^ is enriched with *Sphingomanas*, SM–OF at 1.35 kg/m^2^ is enriched with *GP6*, three types *Acidobacteria_Gp6*, *Planctomycetaceae*, *Planctomycetales*, *Planctomycetia*, *Planctomycetes*, *Rhodospirillales*, CCF at 0.075 kg/m^2^ are enriched with *Actinomycetales*, two types *Actinobacteria*, *Xanthomonadaceae*, and the control is enriched with *Acidobacteria_Gp3*. Furthermore, the difference in relative abundance composition (family level) of the rhizosphere bacterial community under two kinds of commercial organic fertilizers at different concentrations was visually explained through heat maps (Figure 7b). The PMMR–OF treatment was enriched with *WPS–1*, *Xanthomonadaceae*, *Sphingomonadaceae*, *Saccharibacteria*, *Rhodospirillaceae*, *Hyphomicrobiaceae*, *Chitinophagaceae*, and *Acidobacteria_Gp4*, but was reduced with *Acidobacteria_Gp3* and *Spartobacteria* (*p* < 0.05). The SM–OF treatment was enriched with *Spartobacteria* and *Rhodospirillaceae* but was reduced with *WPS–1* and *Chitinophagaceae* (*p* < 0.05).

Meanwhile, LEfSe was also executed at LDA > 4 (*p* < 0.05) in rhizophere soil fungal communities under different treatment (Figure 8a). A total of six biomarkers were found in the PMMR–OF treatments. PMMR–OF at 1.35 kg/m^2^ is rich in *Ramophialophora* and *Sordariales*, and PMMR–OF at 1.80 kg/m^2^ is rich in *Chaetomium*, *Chaetomiaceae*, *Sordariales*, and *Sordariomycetes*. We also further visually explained the difference in the relative abundance composition (family level) of the rhizosphere fungal community under different treatment conditions through heat maps (Figure 8b). The PMMR–OF treatment was enriched with *Ascobolaceae*, *Sordariales*, and *Chaetomiaceae*, but was reduced with *Mortierellaceae*, *Herpotrichiellaceae*, and *Hypocreaceae* (*p* < 0.05). The SM–OF treatment was enriched with *Hypocreaceae* and *Plectosphaerellaceae*, but none were obviously reduced (*p* < 0.05). The results demonstrate that different commercial organic fertilizers such as PMMR–OF and SM–OF can increase or induce the existence of specific species, resulting in a change in the bacterial or fungal community structure in rhizosphere soil of corn. In agreement with these results, previous studies also showed that different organic fertilizers, including commercial organic fertilizer, sheep manure, and mushroom residue, could result in a change in microbial community structure in rhizosphere soil, improving the soil fertility and productivity [4]. In fact, we have isolated four fungal and five bacterial isolates originating from new reclaimed land, which means these isolates may have greater potential to colonize in immature soil. Indeed, these microbes exhibited agreat ability to fix nitrogen, solubilize phosphate, produce siderophores and indole acetic acid, and promote the growth of eggplant [46,54]. In other words, these microbes may be playing an important role in the new reclamation land by altering soil chemistry and microbial communities.

### 3.6. The Effect of Two Kinds of Commercial Organic Fertilizers on RDA of Soil Properties and Mibrobial Communities

Soil properties exhibited an influence in the composition of bacterial and fungal communities in corn rhizosphere soil at the genus levels (Figure 9; Table 6). The association of soil properties with rhizosphere microbial community composition was examined using a redundancy discriminant analysis (RDA), which has been used previously to explore the relationship between environment factors and microbial communities. Results from this study showed that there was a total of 27.47% and 24.62% of the cumulative variance of the rhizosphere microbial community-factor correction at the bacterial (Figure 9a) and fungal (Figure 9b) genus level, respectively. For bacterial communities at the genus level, the contributions of the three main variables were 30.75% by organic matter contents, 24.00% by available K and 20.05% by available P, respectively (Table 6). For fungal communities at the genus level, the contributions of the six main variables were 63.13% by available K, 60.46% by available P, 52.00% by exchangeable Mg, 41.90% by organic matter contents, 27.94% by total N, and 20.43% by pH, respectively (Table 6). In general, the study showed a complex relationship between soil nutrient elements and microbial growth because the compositions of the bacterial and fungal communities in corn rhizosphere soil were significantly affected by different soil properties, and the main soil variables exhibited a greater influence in fungal communities than bacterial communities, which was different from Ren et al. (2021), who thought the soil’s main variables exhibited a great influence in bacterial communities compared to fungal communities [17]. Meanwhile, in agreement with the result of this study, the growth of soil microbe was affected by many environmental factors, such as soil pH and organic matter contents, as well as the content of nitrogen, available P, exchangeable magnesium, and exchangeable calcium [17]. Our previous study also found that the growth of soil microbes was affected by many environmental factors, such as exchangeable Ca, organic matter contents, total N, and available P [3,4]. These results can provide a basis for the application of suitable fertilizers and the production of corn in newly reclaimed land in the future.

## 4. Conclusions

The results of this study showed that two kinds of commercial organic fertilizers (PMMR-OF and SM-OF) did not significantly change the soil pH. However, SM-OF had a better effect on improving soil organic matter contents, total N, available P, available K, and exchangeable Mg. This suggests that SM-OF can be used as a good amendment for the new reclamation land, but the effect of improvement depends on the concentration. In addition, fertilizers caused a differential change in bacterial and fungal community of the new reclamation land. The composition of microbial communities in corn rhizosphere soil was affected significantly by organic matter contents, available P and available K, at the genus level. Some specific microbes, such as *Aureobasidium* (biocontrol fungi), *Mortierella* (plant growth promoting fungi), *Gemmatimonas* (phosphorus solubilizing bacteria), and *Sphingomonas* (plant growth promoting bacteria), have been found to be closely related to the soil improvement by different fertilizers, indicating the relationships among microbes, fertilizer, and soil. Overall, the results of this study revealed significant changes under different fertilizer conditions in microbiota and chemical properties of corn soil. Commercial organic fertilizers, particularly SM-OF, can be used as a good amendment for the new reclamation land.

## Figures and Tables

**Figure 1 plants-11-02553-f001:**
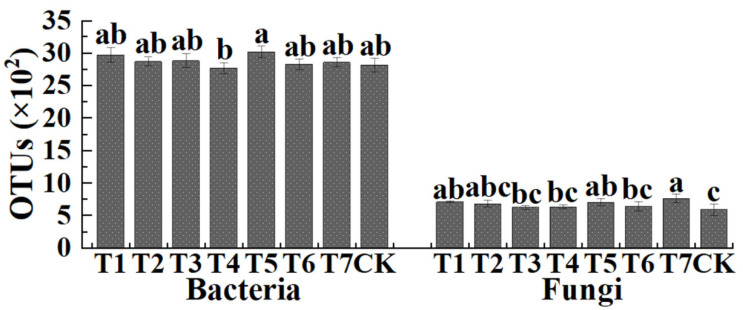
The effect of two kinds of commercial organic fertilizers at different concentrations on the OTU distribution of bacteria and fungi. T1–3: PMMR-OF at 0.90, 1.35, and 1.80 kg/m^2^; T4–6: SM-OF at 0.75, 1.05, and 1.35 kg/m^2^; T7: CCF at 0.075 kg/m^2^; CK: the control. Different lowercase letters reveal the significance among different treatments (*p* < 0.05).

**Figure 2 plants-11-02553-f002:**
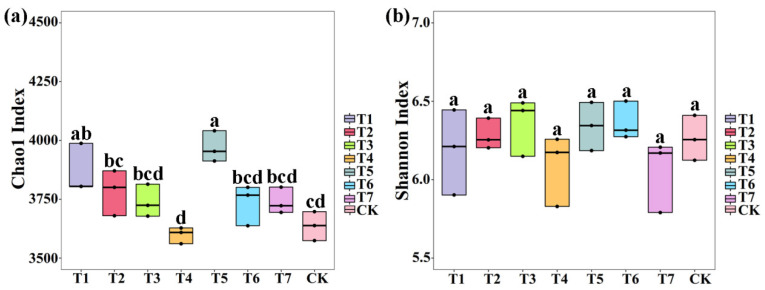
The effect of two kinds of commercial organic fertilizers at different concentrations on Chao1 diversity index (**a**) and Shannon′s diversity index (**b**) of bacteria. T1–3: PMMR-OF at 0.90, 1.35, and 1.80 kg/m^2^; T4–6: SM-OF at 0.75, 1.05, and 1.35 kg/m^2^; T7: CCF at 0.075 kg/m^2^; CK: the control. Different lowercase letters reveal the significance among different treatments (*p* < 0.05).

**Figure 3 plants-11-02553-f003:**
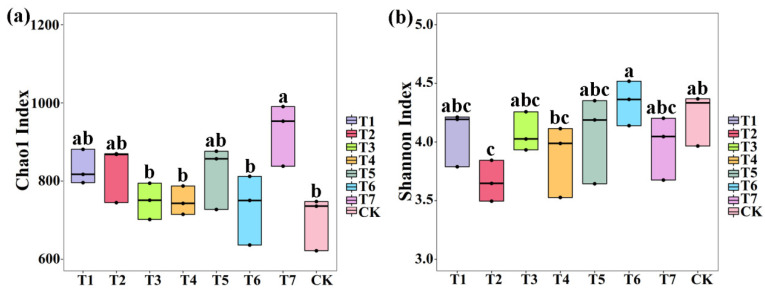
The effect of two kinds of commercial organic fertilizers at different concentrations on Chao1 diversity index (**a**) and Shannon′s diversity index (**b**) of fungi. T1–3: PMMR-OF at 0.90, 1.35, and 1.80 kg/m^2^; T4–6: SM-OF at 0.75, 1.05, and 1.35 kg/m^2^; T7: CCF at 0.075 kg/m^2^; CK: the control. Different lowercase letters reveal the significance among different treatments (*p* < 0.05).

**Figure 4 plants-11-02553-f004:**
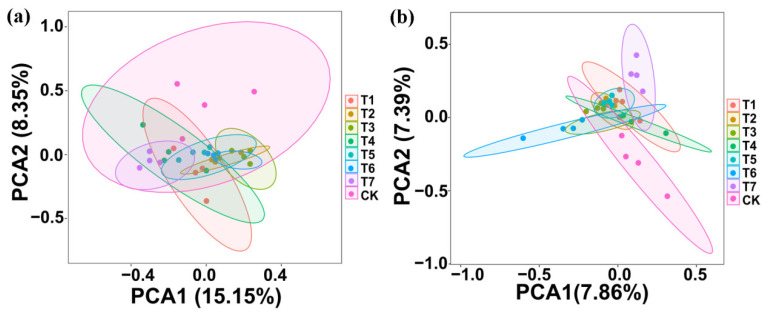
PCA results of soil bacteria (**a**) and fungi (**b**) based on OUT abundance. T1–3: PMMR-OF at 0.90, 1.35, and 1.80 kg/m^2^; T4–6: SM-OF at 0.75, 1.05, and 1.35 kg/m^2^; T7: CCF at 0.075 kg/m^2^; CK: the control.

**Figure 5 plants-11-02553-f005:**
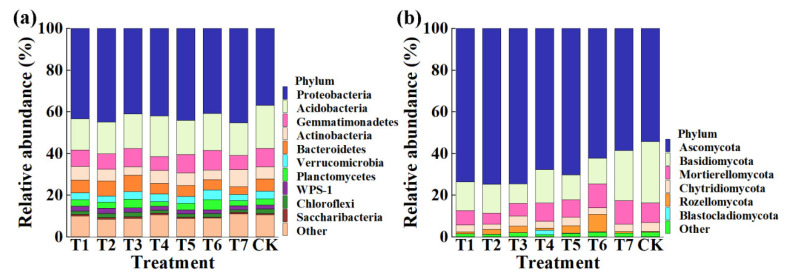
Relative abundance of bacteria (**a**) and fungi (**b**) at the phylum level. T1–3: PMMR-OF at 0.90, 1.35, and 1.80 kg/m^2^; T4–6: SM-OF at 0.75, 1.05, and 1.35 kg/m^2^; T7: CCF at 0.075 kg/m^2^; CK: the control.

**Figure 6 plants-11-02553-f006:**
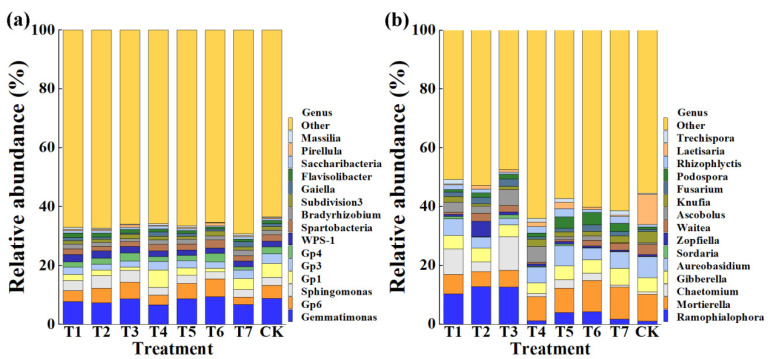
Relative abundance of bacteria (**a**) and fungi (**b**) at the genus level. T1–3: PMMR-OF at 0.90, 1.35, and 1.80 kg/m^2^; T4–6: SM-OF at 0.75, 1.0, 5 and 1.35 kg/m^2^; T7: CCF at 0.075 kg/m^2^; CK: the control.

**Figure 7 plants-11-02553-f007:**
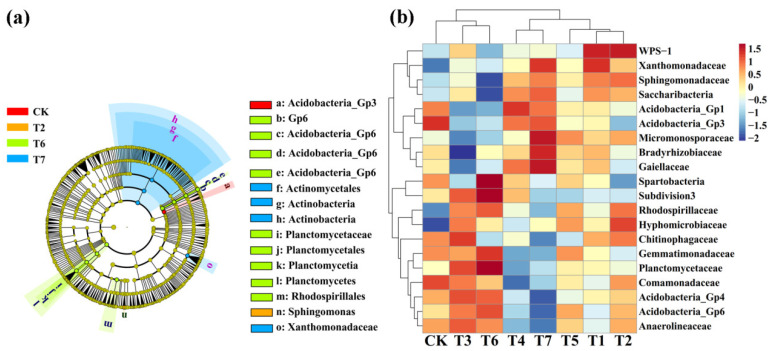
Linear discriminant analysis (LDA) effect size (LEfSe) of the bacterial taxa (**a**), which identifies the most differentially abundant taxa among the two kinds of commercial organic fertilizers at different concentrations treatment. Only taxa with LDA values greater than 4 (*p* < 0.05) are shown. Hierarchical clustering analysis and heat map at the family level (**b**). The tree plot represents a clustering analysis of the top 20 bacteria at family levels according to their Pearson correlation coefficient matrix and relative abundance, and the upper tree plot represents a clustering analysis of soil samples according to the Euclidean distance of data. T1–3: PMMR-OF at 0.90, 1.35, and 1.80 kg/m^2^; T4–6: SM-OF at 0.75, 1.05, and 1.35 kg/m^2^; T7: CCF at 0.075 kg/m^2^; CK: the control.

**Figure 8 plants-11-02553-f008:**
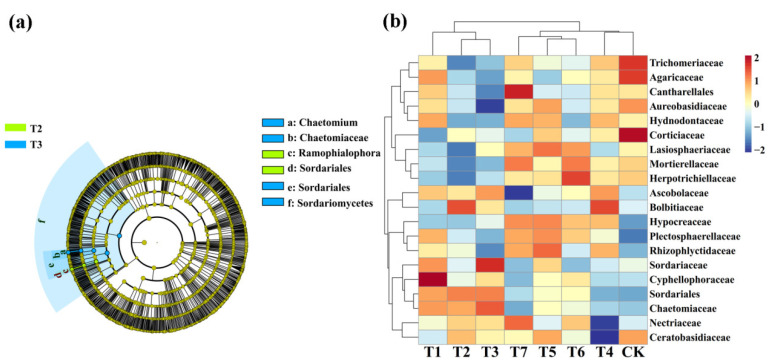
Linear discriminant analysis (LDA) effect size (LEfSe) of the fungal taxa (**a**), which identifies the most differentially abundant taxa among the two kinds of commercial organic fertilizers at different concentrations treatment. Only taxa with LDA values greater than 4 (*p* < 0.05) are shown. Hierarchical clustering analysis and heat map at the family level (**b**). The tree plot represents a clustering analysis of the top 20 fungi at family levels according to their Pearson correlation coefficient matrix and relative abundance, and the upper tree plot represents a clustering analysis of soil samples according to the Euclidean distance of data. T1–3: PMMR-OF at 0.90, 1.35, and 1.80 kg/m^2^; T4–6: SM-OF at 0.75, 1.05, and 1.35 kg/m^2^; T7: CCF at 0.075 kg/m^2^; CK: the control.

**Figure 9 plants-11-02553-f009:**
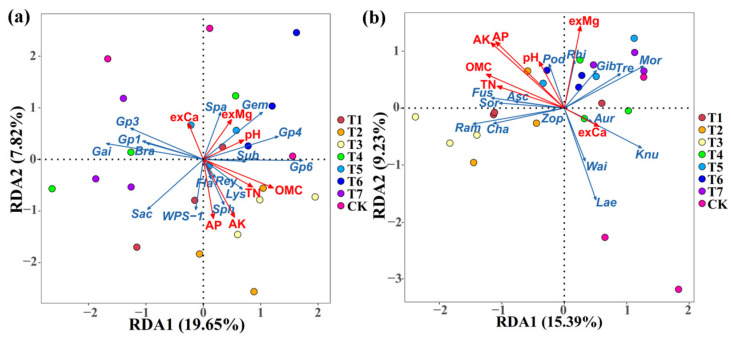
Redundancy discriminant analysis (RDA) of the rhizosphere bacterial (**a**) and fungal (**b**) community compositions at genus levels with soil physicochemical properties. Gem: Gemmatimonas; Sph: Sphingomonas; Spa: Spartobacteria; Bra: Bradyrhizobium; Sub: Subdivision3; Gai: Gaiella; Fla: Flavisolibacter; Sac: Saccharibacteria; Rey: Reyranella; Lys: Lysobacter; Ram: Ramophialophora; Cha: Chaetomium; Gib: Gibberella; Aur: Aureobasidium; Sor: Sordaria; Zop: Zopfiella; Wai: Waitea; Asc: Ascobolus; Knu: Knufia; Fus: Fusarium; Pod: Podospora; Rhi: Rhizophlyctis; Lae: Laetisaria; Tre: Trechispora. OMC: organic matter contents; TN: total N; AP: available P; AK: available K; exCa: exchangeable Ca; exMg: exchangeable Mg. T1–3: PMMR-OF at 0.90, 1.35 and 1.80 kg/m^2^; T4–6: SM-OF at 0.75, 1.05 and 1.35 kg/m^2^; T7: CCF at 0.075 kg/m^2^; CK: the control.

**Table 1 plants-11-02553-t001:** The information of each fertilizer and eight treatments used in this study.

Fertilizers	pH	OMC	NPK Content	Treatments	Concentration (kg/m^2^)
PMMR–OF	8.4	57.4%	8.9%	T1	0.90
				T2	1.35
				T3	1.80
SM–OF	7.8	65.8%	6.5%	T4	0.75
				T5	1.05
				T6	1.35
CCF	4.5		48.0%	T7	0.075
Control				T8	–

PMMR–OF: pig manure and mushroom residue organic fertilizer; SM–OF: sheep manure organic fertilizer; CCF: chemical compound fertilizer; OMC: organic matter conditioner; NPK content: the sum of N, P, K content of each fertilizer T1-3: PMMR-OF at 0.90, 1.35, 1.80 kg/m^2^; T4-6: SM-OF at 0.75, 1.05, 1.35 kg/m^2^; T7: CCF at 0.075 kg/m^2^; T8: the control.

**Table 2 plants-11-02553-t002:** Effects of two kinds of commercial organic fertilizers on corn production.

Treatments (kg/m^2^)	Weight of Air-Dried Corn Ears (kg/plot)					
	2019	GPE%	2020	GPE%	2021	GPE%
PMMR-OF (0.90)	17.27 ± 1.64	12.23 b	20.25 ± 1.98	33.67 a	26.23 ± 1.00	79.90 a
PMMR-OF (1.35)	17.99 ± 0.83	19.17 ab	18.58 ± 0.93	22.67 ab	26.45 ± 0.50	81.38 a
PMMR-OF (1.80)	19.64 ± 1.18	30.12 a	19.56 ± 1.56	29.14 a	25.83 ± 1.31	77.16 a
SM-OF (0.75)	17.29 ± 1.39	14.58 b	17.77 ± 0.86	17.32 ab	26.00 ± 0.73	78.29 a
SM-OF (1.05)	17.43 ± 0.99	15.46 b	17.80 ± 1.48	17.51 ab	26.32 ± 0.49	80.52 a
SM-OF (1.35)	18.66 ± 0.50	23.63 ab	20.09 ± 0.69	32.60 a	23.60 ± 0.52	61.87 b
CCF (0.075)	17.83 ± 1.31	18.11 ab	16.67 ± 0.98	10.03 b	21.21 ± 1.79	45.44 c
Control	15.09 ± 0.56	–	15.15 ± 1.13	–	14.58 ± 0.71	–

PMMR–OF: pig manure and mushroom residue organic fertilizer; SM–OF: sheep manure organic fertilizer; CCF: chemical compound fertilizer. The growth promotion efficacy (GPE%) = (treatment – control)/control × 100%. Data were presented as mean ± standard deviation. Comparison of the means was analyzed using one-way ANOVA followed by Student’s *t*-test. Different lowercase letters within the same columns reveal the significance among different treatments (*p* < 0.05).

**Table 3 plants-11-02553-t003:** Effect of two kinds of commercial organic fertilizers on the pH and organic matter content on new reclamation soil.

Treatments (kg/m^2^)	pH			OMC (g/kg)		
	2019	2020	2021	2019	2020	2021
PMMR-OF (0.90)	5.79 ± 0.27 a	6.40± 0.93 a	6.18± 0.47 a	13.42± 0.32 a	13.95± 1.92 ab	14.77± 1.41 c
PMMR-OF (1.35)	5.76± 0.20 a	6.64± 0.39 a	5.95± 0.43 a	12.72± 1.12 a	15.13± 0.08 ab	16.23± 0.65 abc
PMMR-OF (1.80)	5.76 ± 0.62 a	6.16 ± 0.14 a	5.99± 0.32 a	13.68 ± 0.69 a	16.31 ± 1.27 a	17.14 ± 0.07 ab
SM-OF (0.75)	5.40± 0.17 a	5.88± 0.68 a	6.01 ± 0.21 a	12.21 ± 0.89 ab	13.57 ± 0.97 b	15.87 ± 0.76 bc
SM-OF (1.05)	5.54± 0.12 a	6.13± 0.09 a	6.23 ± 0.15 a	12.41 ± 1.02 ab	14.03 ± 1.20 ab	16.12 ± 0.53 abc
SM-OF (1.35)	5.46± 0.07 a	6.10± 0.17 a	6.28 ± 0.06 a	13.23 ± 0.53 a	16.20 ± 1.35 a	17.67 ± 0.83 a
CCF (0.075)	5.61± 0.42 a	5.92± 0.20 a	6.04 ± 0.33 a	10.53± 1.86 bc	10.57± 1.19 c	11.21± 1.29 d
Control	5.59 ± 0.53 a	5.90 ± 0.26 a	5.92 ± 0.40 a	9.80 ± 0.76 c	10.82 ± 1.03 c	10.08 ± 0.26 d

PMMR–OF: pig manure and mushroom residue organic fertilizer; SM–OF: sheep manure organic fertilizer; CCF: chemical compound fertilizer; OMC: organic matter contents. Data were presented as mean ± standard deviation. Comparison of the means was analyzed using one-way ANOVA followed by Student’s *t*-test. Different lowercase letters within the same columns reveal the significance among different treatments (*p* < 0.05).

**Table 4 plants-11-02553-t004:** Effect of two kinds of commercial organic fertilizers on the total N, available P, and available K on new reclamation soil.

Treatments (kg/m^2^)	Total N (g/kg)			Available P (mg/kg)			Available K (mg/kg)		
	2019	2020	2021	2019	2020	2021	2019	2020	2021
PMMR-OF (0.90)	0.74 ± 0.08 a	0.88 ± 0.06 a	0.99 ± 0.10 a	7.93 ± 0.36 a	12.37 ± 1.05 b	17.80 ± 1.11 ab	438.55 ± 9.38 ab	438.14 ± 8.11 ab	473.62 ± 9.37 ab
PMMR-OF (1.35)	0.73 ± 0.05 a	0.87 ± 0.10 a	1.03 ± 0.21 a	8.03 ± 0.30 a	13.76 ± 0.44 ab	18.40 ± 0.82 ab	420.74 ± 7.39 c	443.27 ± 7.83 a	473.59 ± 6.26 ab
PMMR-OF (1.80)	0.70 ± 0.08 a	0.88 ± 0.08 a	1.06 ± 0.10 a	7.72 ± 0.75 abc	13.64 ± 0.72 ab	18.93 ± 0.69 a	426.88 ± 3.55 bc	444.12 ± 6.47 a	480.04 ± 4.05 a
SM-OF (0.75)	0.74 ± 0.09 a	0.83 ± 0.07 a	1.04 ± 0.16 a	7.11 ± 0.96 cd	12.94 ± 0.98 ab	18.93 ± 0.45 a	376.87 ± 9.41 d	440.72 ± 7.20 ab	467.14 ± 5.02 b
SM-OF (1.05)	0.74 ± 0.04 a	0.89 ± 0.12 a	1.05 ± 0.19 a	7.83 ± 0.69 ab	13.20 ± 1.22 ab	17.53 ± 0.23 ab	414.26 ± 6.03 c	442.79 ± 9.08 a	482.14 ± 6.54 a
SM-OF (1.35)	0.78 ± 0.11 a	0.88 ± 0.13 a	1.06 ± 0.12 a	7.09 ± 0.27 d	13.93 ± 0.75 a	17.43 ± 0.67 b	384.06 ± 6.03 d	426.56 ± 7.77 b	475.58 ± 6.79 ab
CCF (0.075)	0.73 ± 0.05 a	0.74 ± 0.19 a	0.81 ± 0.09 ab	7.29 ± 0.60 bcd	10.87 ± 0.21 c	17.80 ± 0.79 ab	449.67 ± 5.37 a	452.70 ± 9.03 a	449.84 ± 5.53 c
Control	0.64 ± 0.05 a	0.68 ± 0.06 a	0.69 ± 0.05 b	4.64± 0.52 e	4.98 ± 0.55 d	5.28 ± 0.56 c	357.04 ± 8.33 e	371.75 ± 5.80 c	362.02 ± 2.25 d

PMMR–OF: pig manure and mushroom residue organic fertilizer; SM–OF: sheep manure organic fertilizer; CCF: chemical compound fertilizer. Data were presented as mean ± standard deviation. Comparison of the means was analyzed using one-way ANOVA followed by Student’s *t*-test. Different lowercase letters within the same columns reveal the significance among different treatments (*p* < 0.05).

**Table 5 plants-11-02553-t005:** Effect of two kinds of commercial organic fertilizers on the exchangeable Ca and exchangeable Mg on new reclamation soil.

Treatments (kg/m^2^)	Exchangeable Ca (cmol (1/2 Ca^2+^)/kg)			Exchangeable Mg (cmol (1/2 Mg^2+^)/kg)		
	2019	2020	2021	2019	2020	2021
PM-COF (0.90)	16.43 ± 0.74 a	13.93 ± 0.94 ab	9.87 ± 1.12 a	5.73 ± 0.28 a	4.59 ± 0.80 a	4.15 ± 0.15 a
PM-COF (1.35)	14.98 ± 0.55 abc	13.85 ± 1.78 ab	8.78 ± 0.43 ab	5.58 ± 0.10 a	4.57 ± 0.62 a	4.46 ± 0.79 a
PM-COF (1.80)	15.09 ± 0.90 abc	14.29 ± 1.74 a	8.03 ± 0.56 b	5.70 ± 0.05 a	4.88 ± 0.36 a	4.36 ± 0.39 a
SM-COF (0.75)	12.64 ± 1.16 d	11.46 ± 0.53 b	8.68 ± 0.95 ab	5.57 ± 0.41 a	4.42 ± 0.94 a	4.78 ± 0.43 a
SM-COF (1.05)	14.61 ± 1.22 abcd	14.90 ± 0.38 a	9.41 ± 0.60 ab	5.62 ± 0.23 a	4.79 ± 0.34 a	4.97 ± 0.14 a
SM-COF (1.35)	15.23 ± 1.65 ab	13.64 ± 1.27 ab	9.18 ± 0.80 ab	5.78 ± 0.09 a	4.90 ± 0.54 a	5.00 ± 0.01 a
CCF (0.075)	13.87 ± 1.46 bcd	12.43 ± 0.88 ab	8.83 ± 0.31 ab	5.41 ± 0.55 a	4.83 ± 0.60 a	4.66 ± 0.57 a
Control	13.06 ± 0.93 cd	13.62 ± 1.34 ab	9.59 ± 0.51 a	5.44 ± 0.24 a	5.03 ± 0.03 a	4.22 ± 0.99 a

PMMR–OF: pig manure and mushroom residue organic fertilizer; SM–OF: sheep manure organic fertilizer; CCF: chemical compound fertilizer. Data were presented as mean ± standard deviation. Comparison of the means was analyzed using one-way ANOVA followed by Student’s *t*-test. Different lowercase letters within the same columns reveal the significance among different treatments (*p* < 0.05).

**Table 6 plants-11-02553-t006:** Contribution of soil environment to bacteria and fungi taxa at the genus level.

Soil Environment	Contribution at Bacterial Genus Level (%)	Contribution at Fungal Genus Level (%)
pH	10.71	20.43
organic matter contents	30.75	41.90
total N	17.32	27.94
available P	20.05	60.46
available K	24.00	63.13
exchangeable Ca	8.50	9.20
exchangeable Mg	12.85	52.00

## Data Availability

All data supporting the conclusions of this article are included in this article.

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
