# Peer review of "Effects of Two Kinds of Commercial Organic Fertilizers on Growth and Rhizosphere Soil Properties of Corn on New Reclamation Land"

_plants, 2022, doi:10.3390/plants11192553_

Round 1

Reviewer 1 Report

The article „Effects of Two Kinds of Commercial Organic Fertilizers on Growth and Rhizosphere Soil Properties of Corn on New Reclamation Land” presents relevant material on growing corn on reclaimed, often infertile soils. Very detailed research has been done.

Observations:

1. The article presents the influence of fertilizer on corn cultivation and the amount of nutrients in the soil when using different fertilizers. The data presented in the research results are often described with the words "decreased", "increased", but it is not explained why the data values changed. The article would be more interesting to read with this information included.

2. There are layout inconsistencies and inaccuracies. I also suggest editing the first picture.

Author Response

Dear Editor/Reviewers’:

Many thanks for your letter and the reviewer’s comments concerning our manuscript entitled “Effects of Two Kinds of Commercial Organic Fertilizers on Growth and Rhizosphere Soil Properties of Corn on New Reclamation Land” (Manuscript ID: 1906055). Those comments are all valuable and very helpful for revising and improving our paper. We have carefully revised the manuscript according to each comment referred by the reviewers. In accordance with the requirements of the journal, we also have carefully revised the paper format and corrected the grammatical errors. The revised portions were highlighted in the mark-revised manuscript.

Thank you very much for your comments. We appreciate for Editors’/Reviewers’ warm work earnestly, and hope that the correction will meet with approval. Once again thank you very much.

Best wishes for you.

Xuqing Li

Response to Reviewer 1 Comments

The article “Effects of Two Kinds of Commercial Organic Fertilizers on Growth and Rhizosphere Soil Properties of Corn on New Reclamation Land” presents relevant material on growing corn on reclaimed, often infertile soils. Very detailed research has been done.

Response: Many thanks for your acknowledging our research.

Observations:

  1. The article presents the influence of fertilizer on corn cultivation and the amount of nutrients in the soil when using different fertilizers. The data presented in the research results are often described with the words "decreased", "increased", but it is not explained why the data values changed. The article would be more interesting to read with this information included.

Response: Many thanks for your valuable suggestion. Indeed, this manuscript has been revised and improved as suggested by the reviewers.

  1. There are layout inconsistencies and inaccuracies. I also suggest editing the first picture.

Response: Many thanks for your valuable suggestion. We have edited figure 1.

Author Response

Dear Editor/Reviewers’:

Many thanks for your letter and the reviewer’s comments concerning our manuscript entitled “Effects of Two Kinds of Commercial Organic Fertilizers on Growth and Rhizosphere Soil Properties of Corn on New Reclamation Land” (Manuscript ID: 1906055). Those comments are all valuable and very helpful for revising and improving our paper. We have carefully revised the manuscript according to each comment referred by the reviewers. In accordance with the requirements of the journal, we also have carefully revised the paper format and corrected the grammatical errors. The revised portions were highlighted in the mark-revised manuscript.

Thank you very much for your comments. We appreciate for Editors’/Reviewers’ warm work earnestly, and hope that the correction will meet with approval. Once again thank you very much.

Best wishes for you.

Xuqing Li

Response to Reviewer 2 Comments

Comments

The title of the manuscript is: "Effects of Two Kinds of Commercial Organic Fertilizers on Growth and Rhizosphere Soil Properties of Corn on New Reclamation Land”.

Response: OK.

Abstract

  1. Do not start the very first sentence of the manuscript ‘with’, instead start with proper subject. Moreover, the word ‘amount’forland resourcesseems in appropriate here and can find be changed with proportion, or ‘an abundance etc.

Response: Many thanks for the valuable suggestion. We have changed ‘with’ into ‘due to’.

  1. Remove ‘a’ from sentence in line 27.

Response: Many thanks for the valuable suggestion. We have corrected.

  1. The sentence started form line 29 need to be revised and presented in better way as it is too long and not well structured. Need to be divided in few small sentences in precise manner. The significant results need to be elaborated in narrative and precise way that can easily be understanding by non-experts as well.

Response: Many thanks for the valuable suggestion. We have changed the sentence into “Indeed, compared with the control based on16S and ITS amplicon sequencing of soil microflora, SM-OF caused a 10.79-19.52%, 4.33-4.39%, 14.58-29.29% increase in Proteobacteria, Actinobacteria, Ascomycota, but a 5.82-20.58%, 0.53-24.06%, 10.87-16.79%, 2.69-10.50%, 44.90-59.24%, 8.88-10.98%, 2.31-21.98% reduction in Acidobacteria, Gemmatimonadetes, Bacteroidetes, Verrucomicrobia, Basidiomycota, Mortierellomycota, Chytridiomycota, respectively. While CCF caused a 24.11%, 23.28%, 38.87%, 19.88%, 18.28%, 13.89% reduction in Acidobacteria, Gemmatimonadetes, Bacteroidetes, Verrucomicrobia, Basidiomycota, Chytridiomycota, but a 22.77%, 41.28%, 7.88%, 19.39% increase in Proteobacteria, Actinobacteria, Ascomycota, Mortierellomycota, respectively”.

  1. Make it more specific rather giving a general statement ‘different fertilizer conditions’ in line 39.

Response: Many thanks for the valuable suggestion. We have now revised the sentence as suggested.

 Introduction

  1. Revised the sentence in line 62 ‘fertilized largely in a long time’ poorly written and need to be revised.

Response: Many thanks for the valuable suggestion. Modification has been made as suggested through the manuscript. Indeed, we revised ‘fertilized largely in a long time’ into ‘were widely used in a long time’.

  1. Remove ‘Previous research has shown that’ from Line 66.

Response: Many thanks for the valuable suggestion. Modification has been made as suggested.

  1. Line 73-77: The 2nd last sentence is need to be revised ……….”This objective of this study is to evaluate the effect?. Please revise it.

Response: Many thanks for the valuable suggestion. Modification has been made as suggested.

  1. Include your hypothesis and as well as rational for the study conducted.

Response: Many thanks for the valuable suggestion. Modification has been made as suggested.

 Materials and Methods

  1. Replace ‘this’ with ‘this’ experiment in  Line83. The information given in table 1 need to be reduced as it is already explained in the table. Moreover, NPK concentration were combined in Table and it is need to be given separate as it is confusing? What does 8.9% NKP mean? Is it all the three were in same concentration? Which is not possible? Need to be clarified and will be better to gave separate for each nutrient?

Response: Many thanks for the suggestion. We have deleted ‘others’ and ‘price’ in Table . The content of NPK was provided by Fertilizer Company, for example, ‘8.9% NKP in PMMR-OF’ means ‘the sum of N, P, K content in PMMR-OF is 8.9%’. Modification has been made as suggested.

  1. What are T1, T2, T3, etc and concentration given in Table 1? It needs to be given in foot note of the table?

Response: Many thanks for the valuable suggestion. Modification has been made as suggested.

  1. Which testes were used for testing for normality and variance homogeneity.

Response: Many thanks for the valuable suggestion. In this study, student’s t-test was used for testing for normality and variance homogeneity. And we have added the information in each table.

  1. The agronomic data need to be analyzed combine over years as the study was conducted for three years and year might play its role in decomposition of organic residue and may affect major traits in the study. It will be better to give one table for analysis of variance for all traits including treatment and Year and interaction of treatments and Year effects.

Response: This is indeed good advice, which will provide more information. However, we are not good at this kind of statistics analysis, thus, in order to clear this point, we added some related discussion in the revised manuscript.

Results and discussion

  1. This section is better explained however, the results need be more narrative and briefer that may convey the message to audience in easier way. No need to explain everything given in the Tables and Figures.

Response: Many thanks for the valuable suggestion. Modification has been made as suggested.

  1. Figure 1 need to be revised and will be better if constructed in Origin etc. with standareer or bars as well.

Response: Many thanks for the valuable suggestion. Figure 1 have been revised using OriginPro 2021.

  1. Explain values with ± in foot note of each table.

Response: Many thanks for the valuable suggestion. We have now added the annotation in each table.

  1. Reduce the larger paragraphs by splitting in two or more and make it precise and remove repetition.

Response: Many thanks for the valuable suggestion. We have now revised the sentence structure as suggested.

Conclusion

The conclusions section needs to be improved and need to reduced and precise. Only major findings need to be highlighted along with recommendation.  

Finally, I suggest that the manuscript should be carefully checked for English editing because there are many grammar mistakes throughout the manuscript as well as sentences that did not make sense.

Response: Many thanks for the valuable suggestion. Modification has been made as suggested.

Supplementary File Table S1 Effect of two kinds of commercial organic fertilizers on corn production, pH, OMC, total N, available P, available K, exchangeable Ca, exchangeable Mg

Treatments

(kg/m2)

weight of air-dried Corn ears (kg/plot)

pH

OMC

Total N

available P

available K

exchangeable Ca

exchangeable Mg

2019

2020

2021

2019

2020

2021

2019

2020

2021

2019

2020

2021

2019

2020

2021

2019

2020

2021

2019

2020

2021

2019

2020

2021

PM-COF (0.90)

*

*

*

ns

ns

ns

*

*

*

ns

ns

*

*

*

*

*

*

*

*

ns

ns

ns

ns

ns

PM-COF (1.35)

*

*

*

ns

ns

ns

*

*

*

ns

ns

*

*

*

*

*

*

*

*

ns

ns

ns

ns

ns

PM-COF (1.80)

*

*

*

ns

ns

ns

*

*

*

ns

ns

*

*

*

*

*

*

*

*

ns

*↓

ns

ns

ns

SM-COF (0.75)

*

*

*

ns

ns

ns

*

*

*

ns

ns

*

*

*

*

*

*

*

ns

ns

ns

ns

ns

ns

SM-COF (1.05)

*

*

*

ns

ns

ns

*

*

*

ns

ns

*

*

*

*

*

*

*

ns

ns

ns

ns

ns

ns

SM-COF (1.35)

*

*

*

ns

ns

ns

*

*

*

ns

ns

*

*

*

*

*

*

*

*

ns

ns

ns

ns

ns

CCF (0.075)

*

ns

*

ns

ns

ns

ns

ns

ns

ns

ns

ns

*

*

*

*

*

*

ns

ns

ns

ns

ns

ns

Control

-

-

-

-

-

-

-

-

-

-

-

-

-

-

-

-

-

-

-

-

-

-

-

-

PMMR–OF: pig manure and mushroom residue organic fertilizer; SM–OF: sheep manure organic fertilizer; CCF: chemical compound fertilizer. ns: not significant; ↓: reduction; *: significant differences (p < 0.05).

Reviewer 3 Report

In this MS, the authors carried out a series of experiments to study the effects of two Kinds of commercial organic fertilizers (pig manure and mushroom residue organic fertilizer (PMMR-OF) and sheep manure organic fertilizer (SM-OF)) on the growth and rhizosphere soil properties of corn on new reclamation land, the topics of organic fertilizers (vs. chemical compound fertilizer and the control of treatment without any fertilizer) and new reclamation soil (even though no the control of mature soil) are interesting and novel. Moreover, the effects of two kinds of commercial organic fertilizers in soil microbial community structure were also researched. While there were some shortcomings or errors, which were following as:

  Q1: Abstract: L34-36: but a 22.77%, 41.28%, 7.88%, 19.39% increase in ... of soil microflora Not clear about the control treatment in the Abstract.

  Q2: Abstract: L36-38: “redundancy discriminant analysis of .... included organic matter content, available P and available K” Not clear that PMMR-OF or SM-OF compared with CCF or the Control of treatment without any fertilizer.

  Q3: Introduction: L53-55: “Li et al. [4] reported..., the yield of sweet potato is only about 15,000 kg/ha in mature soil” Wrong that 15,000 kg/ha in mature soil. It should be that "Li et al. [4] reported that compared to the normal yield (about 27,600 kg/ha) of sweet potato in mature soil, the yield of sweet potato is only about 15,000 kg/ha in immature or reclamation soil".

  Q4: 2.1. Experimental design: L104-105: “The area of each plot was 20 m2 and the width and depth of the plot were 30 and 20 cm, respectively” It is right that the depth of the plot was 20 cm as upper layer soil, while it may be wrong that the width of the plot was 30 cm, how about the length of the plot to realize that 20 m2 per plot?

  Q5: 2.4. Statistical analysis: How about the ANOVA of the main treatments and their interaction? One-way or twp-way ANOVA? Where was the table of the ANOVAs? Moreover, what about the significant test method used in this study?

  Q6: Results: It was not just compared between PMMR-OF or SM-OF and the control of the treatment without any fertilizer. How about the differences in the measured indexes between PMMR-OF and SM-OF to show the effects of organic fertilizers on corn production and other measured indexes?

  Q7: Table 2: How about the unit? kg per plot or other unite? And what about the significance test method? It should be shown in the table note.

  Q8: The subtitles of 3.3 (The effect of two kinds of commercial organic fertilizers in microbial community structure) and 3.4 (The effect of two kinds of commercial organic fertilizers in soil microbial community structure) were same.

  Q9: Conclusion: Same content (L571-582) in the Conclusion was repeated as those (L29-40) in the Abstract. 

Author Response

Dear Editor/Reviewers’:

Many thanks for your letter and the reviewer’s comments concerning our manuscript entitled “Effects of Two Kinds of Commercial Organic Fertilizers on Growth and Rhizosphere Soil Properties of Corn on New Reclamation Land” (Manuscript ID: 1906055). Those comments are all valuable and very helpful for revising and improving our paper. We have carefully revised the manuscript according to each comment referred by the reviewers. In accordance with the requirements of the journal, we also have carefully revised the paper format and corrected the grammatical errors. The revised portions were highlighted in the mark-revised manuscript.

Thank you very much for your comments. We appreciate for Editors’/Reviewers’ warm work earnestly, and hope that the correction will meet with approval. Once again thank you very much.

Best wishes for you.

Xuqing Li

Response to Reviewer 3 Comments

Comments: In this MS, the authors carried out a series of experiments to study the effects of two Kinds of commercial organic fertilizers (pig manure and mushroom residue organic fertilizer (PMMR-OF) and sheep manure organic fertilizer (SM-OF)) on the growth and rhizosphere soil properties of corn on new reclamation land, the topics of organic fertilizers (vs. chemical compound fertilizer and the control of treatment without any fertilizer) and new reclamation soil (even though no the control of mature soil) are interesting and novel. Moreover, the effects of two kinds of commercial organic fertilizers in soil microbial community structure were also researched. While there were some shortcomings or errors, which were following as:

Response: We are thankful to the reviewer 3 for acknowledging our manuscript and suggesting the changes for further improvement. In addition, the point-by-point response to each comment is listed below.

Q1: Abstract: L34-36: “but a 22.77%, 41.28%, 7.88%, 19.39% increase in ... of soil microflora” Not clear about the control treatment in the Abstract.

Response: Many thanks for your comments. In fact, 22.77%, 41.28%, 7.88%, 19.39% was obtained by comparing CCF with the control ((CCF-control)/control*100).

Q2: Abstract: L36-38: “redundancy discriminant analysis of .... included organic matter content, available P and available K” Not clear that PMMR-OF or SM-OF compared with CCF or the Control of treatment without any fertilizer.

Response: Many thanks for your comments. RDA indicated that organic matter content, available P and available K was the main variables of bacterial and fungal communities for all the treatments including PMMR-OF, SM-OF, CCF and the Control. Modification has been made as suggested through the manuscript.

Q3: Introduction: L53-55: “Li et al. [4] reported..., the yield of sweet potato is only about 15,000 kg/ha in mature soil” Wrong that 15,000 kg/ha in mature soil. It should be that "Li et al. [4] reported that compared to the normal yield (about 27,600 kg/ha) of sweet potato in mature soil, the yield of sweet potato is only about 15,000 kg/ha in immature or reclamation soil".

Response: Many thanks for your carefulness. We have now corrected as reviewer 3 suggested, and changed the former formulation into "Li et al. [4] reported that compared to the normal yield (about 27,600 kg/ha) of sweet potato in mature soil, the yield of sweet potato is only about 15,000 kg/ha in reclamation soil".

Q4: 2.1. Experimental design: L104-105: “The area of each plot was 20 m2 and the width and depth of the plot were 30 and 20 cm, respectively” It is right that the depth of the plot was 20 cm as upper layer soil, while it may be wrong that the width of the plot was 30 cm, how about the length of the plot to realize that 20 m2 per plot?

Response: Many thanks for your carefulness. The area of each plot was 20 m2, and the width, length and depth of the plot were 330, 600 and 20 cm, respectively. And the planting density of corn is 30 cm * 20 cm.

Q5: 2.4. Statistical analysis: How about the ANOVA of the main treatments and their interaction? One-way or twp-way ANOVA? Where was the table of the ANOVAs? Moreover, what about the significant test method used in this study?

Response: Many thanks for the valuable suggestion. One-way ANOVA used in this study. Modification has been made as suggested through the manuscript.

Q6: Results: It was not just compared between PMMR-OF or SM-OF and the control of the treatment without any fertilizer. How about the differences in the measured indexes between PMMR-OF and SM-OF to show the effects of organic fertilizers on corn production and other measured indexes?

Response: Many thanks for the valuable suggestion. Modification has been made as suggested through the manuscript.

Q7: Table 2: How about the unit? kg per plot or other unite? And what about the significance test method? It should be shown in the table note.

Response: Many thanks for your carefulness. It should be kg/plot. One-way ANOVA and Student’s t-test is the significance test method. Modification has been made as suggested through the manuscript.

Q8: The subtitles of 3.3 (The effect of two kinds of commercial organic fertilizers in microbial community structure) and 3.4 (The effect of two kinds of commercial organic fertilizers in soil microbial community structure) were same.

Response: Many thanks for your comments. In fact, in our manuscript, 3.3 (The effect of two kinds of commercial organic fertilizers in microbial community diversity) and 3.4 (The effect of two kinds of commercial organic fertilizers in soil microbial community structure) were different, please check, thanks.

Q9: Conclusion: Same content (L571-582) in the Conclusion was repeated as those (L29-40) in the Abstract. 

Response: Thanks for the valuable suggestion. Modification has been made as suggested through the manuscript.

Round 2

Reviewer 3 Report

This version has been revised based on the review comments.